# LANGUAGE MODELS DO NOT HAVE HUMAN-LIKE WORKING MEMORY

## ABSTRACT

While Large Language Models (LLMs) exhibit remarkable reasoning abilities, we demonstrate that they lack a fundamental aspect of human cognition: working memory. Human working memory is an active cognitive system that enables not only the temporary storage of information but also its processing and utilization, enabling coherent reasoning and decision-making. Without working memory, individuals may produce unrealistic responses, exhibit self-contradictions, and struggle with tasks that require mental reasoning. Existing evaluations using N-back or context-dependent tasks fall short as they allow LLMs to exploit external context rather than retaining the reasoning process in the latent space. We introduce three novel tasks: (1) Number Guessing, (2) Yes-No Deduction, and (3) Math Magic, designed to isolate internal representation from external context. Across seventeen frontier models spanning four major model families, we consistently observe irrational or contradictory behaviors, indicating LLMs' inability to retain and manipulate latent information. Our work establishes a new benchmark for evaluating working memory in LLMs and highlights this limitation as a key bottleneck for advancing reliable reasoning systems. Code and prompts will be made publicly available upon publication.

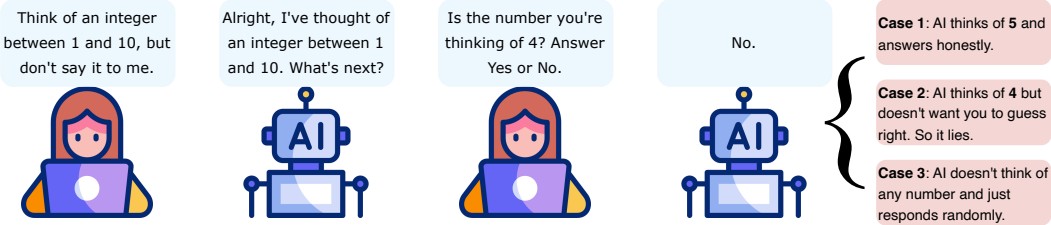

Figure 1: When LLMs say they already have a number in mind, and it is not 4, how can we know whether LLMs are lying, or even thinking of nothing?

## 1 INTRODUCTION

A notable feature of modern Large Language Models (LLMs) is their ability to perform tasks across a wide range of domains, including law (Guha et al., 2023), education (Wen et al., 2024), translation (Jiao et al., 2023), and healthcare (Yang et al., 2024b). Most evaluations focus on their extrinsic behaviors—what tasks they can and cannot perform. To better understand fundamental, underlying model capabilities, a growing body of work examines their intrinsic behaviors, core abilities that shape the downstream task performance. Inspired by cognitive science, recent studies examine whether LLMs exhibit human-like features such as personality (Huang et al., 2024b), emotion (Huang et al., 2024a), empathy (Sorin et al., 2024), theory-of-mind (Liu et al., 2024b), and values (Wang et al., 2024b).

One such ability is memory, which has attracted increasing attention from both industry and research communities. OpenAI was the first to introduce a memory module in ChatGPT (February 2024)[1] that allowed the model to remember information from previous interactions with a user, such as the

---

[1]https://openai.com/index/memory-and-new-controls-for-chatgpt/

user's facts and preferences. The model could access and retrieve these "memories" to be used in later conversations. By mid-2025, xAI, Anthropic, and Google have integrated memory into Grok,[2] Claude,[3] and Gemini,[4] respectively.

**What is memory?** Atkinson & Shiffrin (1968) categorized memory by retention timescale: sensory (1ms–2s), short-term (seconds), and long-term (hours to lifetime). Long-term memory enables information storage over extended periods, whereas short-term, or *Working Memory*, maintains and manipulates information during complex tasks such as reasoning, comprehension, and learning (Baddeley & Hitch, 1974). These distinctions have also been adopted in machine learning, corresponding to representational learning for raw inputs (sensory memory), in-context computation at test time (working memory), and access to external databases (long-term memory) (Weng, 2023). Researchers in the AI community have investigated long-term memory mechanisms for both individual LLMs (Wu et al., 2025; Du et al., 2025) and LLM agents (Zhang et al., 2025; Xu et al., 2025). Recent work has shifted from context-dependent approaches (*e.g.*, Chain-of-Thought (CoT) (Wei et al., 2022) and scratchpads (Lanchantin et al., 2023)) to explicit storage methods (*e.g.*, text-based (Park et al., 2023) or vector-based (Hatalis et al., 2023)) to support lifelong memory (Zheng et al., 2025; Wang et al., 2025b). Most studies frame memory as an engineering problem: enabling LLMs to store and retrieve information for later use.

By contrast, **working memory remains relatively underexplored**. Existing studies often adopt the N-back task (Kirchner, 1958) to assess LLMs' working memory (Gong et al., 2024; Zhang et al., 2024). Yet, a fundamental limitation arises: the critical information for correct responses remains accessible in the model's input context, allowing models to "look back" rather than actively maintain internal state. Therefore, these tests are exploring aspects of the context window, not working memory directly. Unlike humans, who cannot revisit prior steps explicitly, LLMs can simply attend to earlier tokens within their context window. Fig. 2 illustrates this discrepancy. To more faithfully evaluate working memory, it is necessary to design experiments where the key information is not explicitly present in the context and is only available if stored in the working memory of the model. Imagine the following scenario: You select a number between one and ten. When ready, you are asked, "Is the number greater than five?" If you answer, observers can reasonably infer that the number has entered your conscious awareness (*i.e.*, your working memory), since clear perception is necessary to perform the comparison and provide a response.[5]

We ask: **Do LLMs possess human-like working memory, or do they only appear to reason by exploiting their context window?** LLMs are often viewed as reasoning in two modes: (1) the *token space* over sequences (Wei et al., 2022; Yao et al., 2023), and (2) the *latent space* over activations (Hao et al., 2024; Geiping et al., 2025). We argue that working memory is necessary to enable stronger latent space reasoning, as the model does not have access to its external reasoning tokens. Evaluating working memory provides insight into whether models can hold and manipulate latent concepts without explicit externalization. Success in this capacity could enhance reasoning without reliance on CoT, as it directly tests the model's ability to maintain objects and concepts internally. Conversely, deficits in working memory impair information processing in humans (Gruszka & Nkecka, 2017; Cowan, 2014), and in LLMs manifest as unrealistic outputs, self-contradictions, and failures on tasks requiring mental manipulation.

The central challenge in designing such evaluations is: **How can we demonstrate the presence or absence of internal memory when we cannot directly observe a model's mind?** To address this limitation, we design three experiments—(1) Number Guessing, (2) Yes–No Deduction, and (3) Math Magic—that test whether LLMs can internally maintain information that is not explicitly present in the input context. Our experiments span 17 frontier LLMs, both proprietary and open-source, including multiple model families and reasoning approaches. Across all settings, the results converge: current LLMs show little evidence of intrinsic working memory. Instead, their reasoning appears to depend on externalized context. These findings suggest that progress in reasoning will require not only larger models or better prompting, but also closer attention to the mechanisms that could endow LLMs with genuine working memory.

---

[2]https://x.com/grok/status/1912670182012801156
[3]https://www.anthropic.com/news/claude-4
[4]https://blog.google/products/gemini/temporary-chats-privacy-controls/
[5]A possibility remains that your response was given by chance if you tell a lie (Fig. 1 Case 2) or do not think of a number at all (Fig. 1 Case 3).

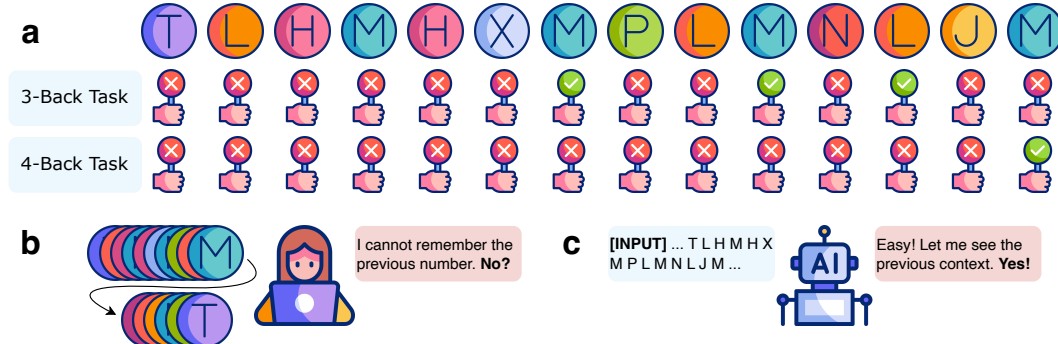

Figure 2: **a.** An illustration of how "N-Back" tasks are performed. **b.** Humans see the stimuli one after one, forcing them to put the information in working memory. **c.** Researchers put all stimuli into context, enabling LLMs to easily find the answers.

## 2 PRELIMINARIES

### 2.1 HOW WORKING MEMORY IS EVALUATED

Human working memory is typically assessed using behavioral paradigms that require individuals to maintain and update information over short intervals. Common examples include the *digit span task* (Miller, 1956), where participants recall sequences of numbers of increasing length, and the *N-back task* (Kirchner, 1958), which requires identifying whether the current stimulus matches one presented $N$ steps earlier. These tasks are widely used because they probe the ability to maintain and manipulate information that is no longer externally visible, thereby capturing the essence of working memory function. In the context of LLMs, working memory has been used more loosely: most studies use the term to describe an LLM's capacity to process information within a fixed context window (Li et al., 2023; Guo et al., 2023). Gong et al. (2024); Zhang et al. (2024) evaluate LLMs using N-back tasks. However, as noted in §1, such human-designed tests are not directly valid for LLMs, since models can simply attend to retained tokens in their context window without actively maintaining information internally.

### 2.2 EXISTING ENGINEERING SOLUTIONS

A parallel line of work introduces explicit engineering solutions by equipping LLMs with external memory modules (Hu et al., 2025; Zeng et al., 2024). For example, Wang et al. (2024a) incorporate symbolic working memory to enhance reasoning, while Kang et al. (2024) use it to improve training efficiency. Other approaches implement scratch spaces (Lanchantin et al., 2023), internal CoT mechanisms (Jaech et al., 2024; OpenAI, 2025), and external vector databases (Hatalis et al., 2023), which may partially mitigate working memory limitations identified in this paper. However, these methods do not address whether LLMs possess an intrinsic working memory capacity, analogous to humans' ability to mentally simulate and manipulate objects. The distinction can be illustrated with an analogy: calculators make arithmetic trivial, yet schools continue to assess addition, subtraction, multiplication, and division. The purpose is not solely to solve problems efficiently, but to reveal individuals' underlying cognitive abilities. Similarly, while engineering techniques can extend an LLM's effective memory, we are ultimately interested in **whether the model intrinsically has the basic cognitive capability of working memory**. Without it, a model may function adequately through external tools, but its intrinsic reasoning ability remains fundamentally limited.

### 2.3 EXPERIMENTAL DESIGN

All three experiments in this paper are designed to evaluate whether LLMs exhibit human-like behaviors where an effective working memory is presented, as evidenced by their ability to perform our proposed tasks. Since we cannot let models reveal what they are privately thinking, the three experiments each test a different hypothesis, corresponding to distinct consequences of impaired working memory. (1) The Number Guessing game (§3) evaluates whether an LLM's response distribution

across repeated identical queries remains valid. (2) The Yes–No Deduction game (§4) examines whether LLMs contradict themselves. (3) The Math Magic (§5) assesses whether LLMs' internal reasoning produces correct outcomes. We evaluate 17 frontier LLMs, including GPT (4o (Hurst et al., 2024) and 4o-Mini (OpenAI, 2024)), o-series (o1-Mini (Jaech et al., 2024), o3-Mini, and o4-Mini (OpenAI, 2025)), LLaMA (3.1 (Meta, 2024) and 3.3), Qwen-2.5 (Yang et al., 2024a) (7B and 72B), QwQ (Team, 2024), and DeepSeek (V3 (Liu et al., 2024a) and R1 (Guo et al., 2025)). All models are configured with a temperature of $1.0$ and a top-p value of $1.0$.

## 3 NUMBER GUESSING

**Hypothesis.** Consider a number-guessing game in which a human participant privately selects an integer between one and ten. The experimenter then asks whether the chosen number is one. By repeating this procedure multiple times, we can estimate the probability of selecting one, denoted $p_1$. Extending this process to other numbers yields an estimated distribution over all choices, denoted $p_1, \ldots, p_n$. It is worth noting that, **if the participant truly selects a number and responds honestly, the estimated probabilities should form a valid distribution**, satisfying $\sum_{i=1}^{n} p_i = 1$. In contrast, if an LLM does not base its responses on an actual hidden choice, the resulting estimates will typically violate this constraint, producing $\sum_{i=1}^{n} p_i \neq 1$.

**Setup.** Leveraging this hypothesis, we design a controlled experiment. In each trial, the model is given a fixed prompt: "USER: *Think of an integer between 1 and 10, but don't say it to me.* ASSISTANT: *Got it! I've thought of an integer between 1 and 10. What's next?*" The model is then independently prompted 200 times for $i = 1, \ldots, 10$ with queries such as "*Is the number you're thinking of $i$? Answer Yes or No.*" We record the frequency of "Yes" responses for each number and compute the estimated probabilities $p_i$. If the sum of these probabilities deviates significantly from one, it suggests that the LLM either is not maintaining a number commitment or lies to users.

**Results.** Fig. 3 presents the probabilities of "Yes" responses in each model for numbers from one to ten. Two key observations emerge: (1) **Most LLMs never produce a "Yes" response**; "No" dominates across models. This produces invalid distributions, further indicating that models are estimating the probability of a human guess being correct (10% in our setting, typically very low) rather than maintaining a private number choice. Given that LLMs generally follow instructions and do not deliberately deceive, we attribute this behavior to their failure to internally "think of" a number. (2) When LLMs do respond affirmatively, they exhibit a pronounced **preference for the number seven**. This tendency mirrors human biases (Miller, 1956; Kubovy & Psotka, 1976).

We quantify LLM performance on this task using the sum of probabilities. A value closer to one indicates better model performance. Table 1 reports these sums for each model.

Table 1: The sum of probabilities of each model responding "Yes" for all numbers from one to ten. Color intensity reflects proximity to one: red indicates values closer to zero, while blue signifies values greater than one.

| Model | Sum |
|---|---|
| GPT-4o-Mini-2024-07-18 | 0 |
| GPT-4o-2024-05-13 | 0 |
| GPT-4o-2024-08-06 | 1.085 |
| GPT-4o-2024-11-20 | 0 |
| GPT-4.1-2025-04-14 | 0 |
| o1-Mini-2024-09-12 | 0.005 |
| o3-Mini-2025-01-31 | 0.205 |
| o4-Mini-2025-04-16 | 0.030 |
| LLaMA-3.3-70B-Instruct-Turbo | 0.045 |
| LLaMA-3.1-8B-Instruct-Turbo | 0.980 |
| LLaMA-3.1-70B-Instruct-Turbo | 0.465 |
| LLaMA-3.1-405B-Instruct-Turbo | 1.195 |
| Qwen2.5-7B-Instruct-Turbo | 0 |
| Qwen2.5-72B-Instruct-Turbo | 0 |
| QwQ-32B | 0.005 |
| DeepSeek-V3 | 0 |
| DeepSeek-R1 | 0.640 |

Several observations stand out: (1) Newer models do not necessarily outperform older ones. Within the GPT family, the 0806 version of GPT-4o (the model currently served under the "gpt-4o" API) achieves the best performance, surpassing both the 1120 version and GPT-4.1. Similarly, LLaMA-3.3 underperforms relative to LLaMA-3.1. (2) Using CoT reasoning does not improve performance. Models employing such strategies—o1, o3, o4, QwQ, and DeepSeek-R1—fail to produce probability sums closer to one. (3) Overall, LLaMA-3.1 performs

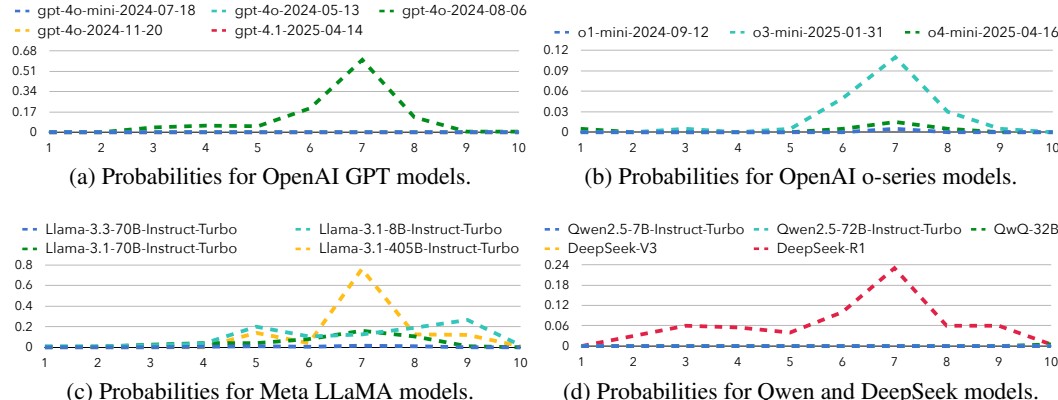

(a) Probabilities for OpenAI GPT models.   (b) Probabilities for OpenAI o-series models.

(c) Probabilities for Meta LLaMA models.   (d) Probabilities for Qwen and DeepSeek models.

Figure 3: Probabilities of model answering "Yes" for each number from one to ten.

best, with the 8B variant outperforming both the 405B and 70B versions. Taken together, these findings suggest that acquisition of this capability appears largely stochastic and is less predictable with respect to model scale. More broadly, they suggest that the observed memory limitations arise not from model size or training sophistication but from a fundamental architectural deficiency.

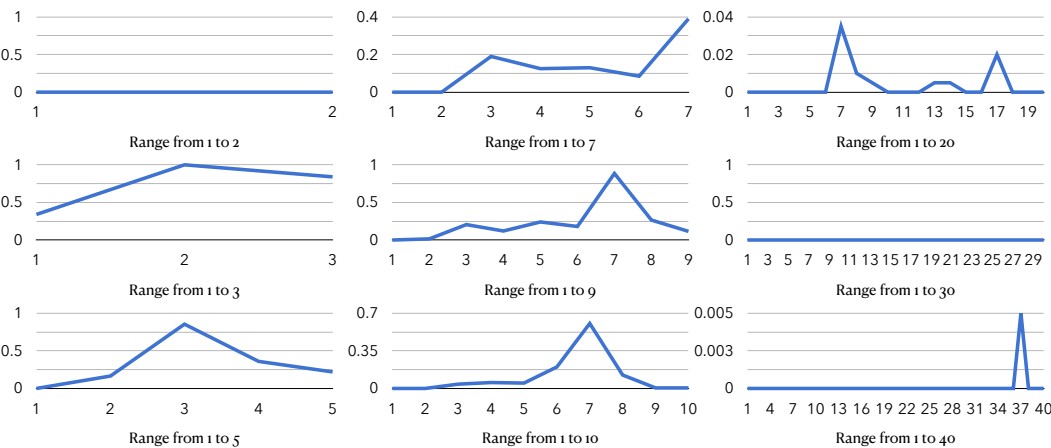

Figure 4: Probabilities of GPT-4o-2024-08-06 answering "Yes" for each number in different ranges.

We further extend our experiments to include a broader range of numbers. Given that GPT-4o-2024-08-06 performs best among the OpenAI models, we focus on its behavior across different numerical ranges. Table 2 reports the summed probabilities for each range, while Fig. 4 illustrates the probability of individual numbers. Our findings reveal two key patterns: (1) For smaller ranges such as 3, 5, and 9, the model exhibits a strong bias toward answering "Yes," with the probability sum significantly exceeding one. In contrast, for larger ranges like 20, 30, and 40, "Yes" responses are rare. (2) When the model does produce a "Yes" response, it frequently corresponds to numbers ending in seven (*e.g.*, 7, 17, 37), as shown in Fig. 4.

Table 2: The sum of probabilities of GPT-4o-2024-08-06 responding "Yes" for all numbers in different ranges.

| Number | Sum |
|--------|-------|
| 2 | 0 |
| 3 | 2.180 |
| 5 | 1.600 |
| 7 | 0.920 |
| 9 | 2.025 |
| 10 | 1.085 |
| 20 | 0.080 |
| 30 | 0 |
| 40 | 0.005 |

**Ablation.** To assess whether our findings are merely artifacts of the decoding scheme, we conduct additional experiments varying temperature and top-p while keeping the other parameter fixed. We use GPT-4o-2024-08-06, the model whose probability sum is closest to one among all GPT-4o variants. The results are summarized in Table 3. Across temperatures of {0.1, 0.4, 0.7, 1.0},

Table 3: Ablation study on decoding parameters: temperature and top-p.

| Settings | 1 | 2 | 3 | 4 | 5 | 6 | 7 | 8 | 9 | 10 | Sum |
|---|---|---|---|---|---|---|---|---|---|---|---|
| T1.0 P1.0 | 0.0 | 0.0 | 0.04 | 0.055 | 0.05 | 0.2 | 0.605 | 0.125 | 0.005 | 0.005 | 1.085 |
| T0.7 P1.0 | 0.0 | 0.005 | 0.01 | 0.09 | 0.055 | 0.225 | 0.765 | 0.09 | 0.005 | 0.0 | 1.245 |
| T0.4 P1.0 | 0.0 | 0.0 | 0.0 | 0.025 | 0.005 | 0.165 | 0.86 | 0.04 | 0.0 | 0.0 | 1.095 |
| T0.1 P1.0 | 0.0 | 0.0 | 0.0 | 0.0 | 0.0 | 0.085 | 0.895 | 0.005 | 0.0 | 0.0 | 0.985 |
| T1.0 P0.7 | 0.0 | 0.0 | 0.0 | 0.0 | 0.0 | 0.1 | 0.83 | 0.02 | 0.0 | 0.0 | 0.95 |
| T1.0 P0.4 | 0.0 | 0.0 | 0.0 | 0.0 | 0.0 | 0.03 | 0.825 | 0.0 | 0.0 | 0.0 | 0.855 |
| T1.0 P0.1 | 0.0 | 0.0 | 0.0 | 0.0 | 0.0 | 0.02 | 0.82 | 0.0 | 0.0 | 0.0 | 0.84 |

the probability sum remains close to one, although lower temperatures concentrate the distribution even more heavily on the number 7. For top-p of $\{0.1, 0.4, 0.7, 1.0\}$, the probability sum decreases as top-p becomes smaller. Despite these variations, the qualitative behavior remains unchanged: the model does not produce a valid probability distribution (*i.e.*, the sum does not converge to one except by coincidence), and its predictions continue to exhibit a strong bias toward 7. This indicates that the failure to commit to a latent number is not driven by the decoding mechanism but reflects a more fundamental limitation in internal state maintenance.

In conclusion, LLMs fail to generate distributions consistent with internally committing to a number. Their outputs are either dominated by "No" response or reflect biased heuristics towards seven. These findings suggest that LLMs struggle to represent and sustain latent numerical values without explicit contextual grounding, thereby highlighting a gap in working memory-like capacity.

## 4 YES-NO DEDUCTION

**Hypothesis.** "Yes-No" (or the Twenty questions[6]) is a social deduction game commonly used to train human reasoning, classification, and questioning skills. In this game, one player privately selects an object, while the opponent asks yes–no questions (*e.g.*, "Is the object heavier than an elephant?") to progressively narrow down the possibilities and ultimately find the object. Consider the decision-making process of the player answering questions, each question requires simply direct comparison between the imagined object and the queried attribute. Note that humans typically do not recall all previous questions and answers. Instead, **they rely on the single reference of the object, without checking self-contradiction with all prior responses**.

We hypothesize that if LLMs cannot maintain such an imagined object in working memory, they can only respond to questions by checking consistency with their prior answers. As the number of questions increases, maintaining consistency becomes increasingly difficult, making the task strongly dependent on long-context reasoning. To test this, we first instruct the LLM to imagine an object and answer a sequence of comparative questions against the reference objects. The goal is to assess whether the model produces self-contradictions. For instance, the model might initially answer "Yes" to "*Is the object heavier than an elephant?*" but later also respond "Yes" to "*Is the object lighter than a cat?*", thereby contradicting itself.

**Setup.** We predefine five sets of objects that are commonly regarded as comparable with respect to five properties: volume, length, weight, density, and hardness. In total, 60 distinct objects are included, as listed in Table 4, ordered by the corresponding property. For each question, one property is randomly selected, followed by an object from the corresponding object list. The model is then prompted to assess whether the object it imagined is *comparative* relative to the given object, where the *comparative* form can vary in direction (*e.g.*, bigger or smaller for volume). In each trial, the model is continuously presented with up to 250 such questions. We record the number of questions it completed before the model exhibits a self-contradiction. If no contradiction is observed across all 250 questions, the trial is considered a *Pass*. Each model is tested with 200 trials.

**Results.** Table 5a presents the number of failed trials for the GPT-4o-2024-08-06 (Hurst et al., 2024) and GPT-4o-Mini-2024-07-18 (OpenAI, 2024). The smaller model (GPT-4o-Mini) consis-

---

[6]https://en.wikipedia.org/wiki/Twenty_questions

Table 4: Objects ordered by the five properties (smallest to largest).

| Volume | Length | Weight | Density |
|---|---|---|---|
| Coffee bean | Rice | Coin | Air |
| Dice | Paperclip | Spoon | Wood |
| Golf ball | Credit card | Watch | Ice |
| Soda can | Pencil | Smartphone | Water |
| Soccer ball | Laptop | Bottle of water | Plastic |
| Microwave oven | Baseball bat | Dictionary | Glass |
| Washing machine | Guitar | Cat | Iron |
| Bathtub | Door | Bicycle | Copper |
| Car | Apple tree | Television | Silver |
| School bus | Coconut tree | Refrigerator | Gold |
| Shipping container | Tennis court | Tiger | **Hardness** |
| Olympic swimming pool | Swimming pool | Cow | Marshmallow |
| Boeing 747 | Football field | Rhino | Rubber eraser |
| Titanic | Skyscraper | Elephant | Brick |
| Great Pyramid of Giza | Mount Everest | Train | Hammer |
| | | | Diamond ring |

Table 5: Count of failures of Yes-No Deduction on the five properties.

(a) GPT-4o-Mini-2024-07-18 and GPT-4o-2024-08-06.

| Model | Failure | V | W | L | D | H |
|---|---|---|---|---|---|---|
| GPT-4o-Mini | 200 | 12 | 46 | 49 | 52 | 41 |
| GPT-4o | 173 | 21 | 42 | 57 | 27 | 26 |

(b) Ablation studies using GPT-4o-2024-08-06.

| Model | Failure | V | W | L | D | H |
|---|---|---|---|---|---|---|
| Hint | 194 | 37 | 39 | 60 | 37 | 21 |
| All | 158 | 18 | 29 | 21 | 55 | 35 |
| Hint + All | 145 | 13 | 32 | 34 | 46 | 20 |

tently fails, while the larger GPT-4o successfully passes 27 out of 200 trials. This result supports our hypothesis: **model performance on this task depends on their long-context processing ability rather than intrinsic working memory for maintaining imagined objects**.

Figure 5 presents histograms of the number of questions each model completes before exhibiting self-contradiction. The distribution for GPT-4o-Mini peaks in the 20–30 range, whereas GPT-4o peaks in the 30–40 range. Moreover, GPT-4o demonstrates a higher frequency of completions in the 80–130 range compared to GPT-4o-Mini. Notably, the types of properties that lead to self-contradictions differ between the two models: GPT-4o-Mini fails more frequently on density and hardness, while GPT-4o shows greater robustness on these attributes.

**Ablation.** To ensure that the observed failures are not simply due to LLMs' inability to rank objects by the five properties (*i.e.*, the lack of commonsense knowledge about object properties), we conduct the following ablation studies: **(1) Hint**: At the beginning of the prompt, we provide GPT-4o with the object rankings defined in Table 4. **(2) All**: For each question, we specify the target object $\mathcal{O}$ by stating explicitly that "the object you are considering is $\mathcal{O}$." **(3) Hint + All**: We combine the above two settings. Results are shown in Table 5b. Two key findings emerge: (1) Providing hints does not prevent contradictions, indicating that the task depends more on long-context reasoning rather than factual knowledge. (2) Explicitly specifying the object substantially reduces errors, effectively collapsing the long-context reasoning task into a short-context reasoning problem.

Across the conditions, models exhibit self-contradictions (*e.g.*, claiming an object is both larger than a car and smaller than a soccer ball) as the number of queries increases. This behavior suggests their reliance on long-context reasoning rather than possessing a dedicated working memory for maintaining such an internal state.

## 5 MATH MAGIC

**Hypothesis.** Consider the following recreational arithmetic game, a variant of the Kaprekar routine (Kaprekar, 1955), which relies on digit manipulation in base 10. Think of a three-digit number in which the hundreds and units digits differ (*e.g.*, *abc*). Reverse the digits to form a new number

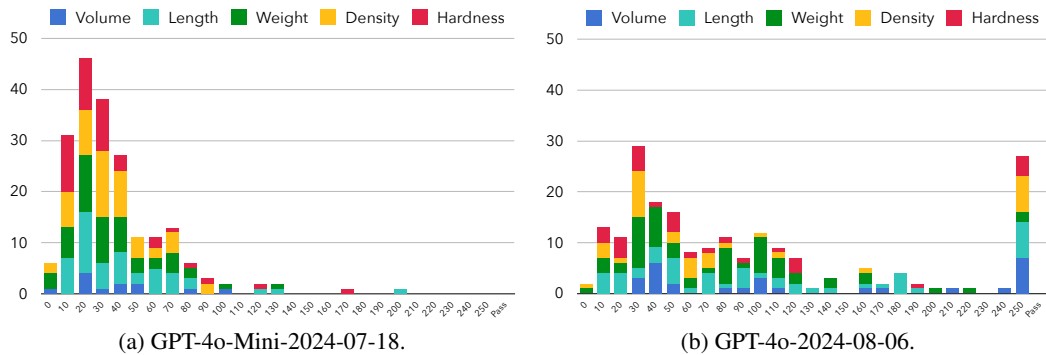

(a) GPT-4o-Mini-2024-07-18.  (b) GPT-4o-2024-08-06.

Figure 5: The histogram of the number of questions where the two models show self-contradiction.

Table 6: Operations in the math magic in our experiment. the random number $a$ ranges from 1 to 7, while the random numbers $b$ and $c$ range from 1 to 3.

| Role | Content |
|------|---------|
| User | Think of 4 integers between 1 and {NUMBER} in order, but don't tell me. |
| Assistant | Okay! I've got 4 numbers. What's next? |
| User | In order, append the same 4 numbers after the original ones. |
| Assistant | Understood! Now I have 8 numbers. What's next? |
| User | Move the first {random_number_a} numbers to the end. |
| Assistant | Got it! Now I have moved the numbers. What's next? |
| User | Take the first 3 numbers and insert them anywhere in the middle. |
| Assistant | Okay! The first 3 numbers are placed somewhere in the middle. What's next? |
| User | Set the first number aside. We don't need it for now. |
| Assistant | Understood! Now I have 7 numbers. What's next? |
| User | Take the first {random_number_b} numbers and insert them anywhere in the middle. |
| Assistant | Got it. The first {random_number_b} numbers are placed somewhere in the middle. What's next? |
| User | Remove the first {random_number_c} numbers. We will never need it anymore. |
| Assistant | Okay! Now I have {7 - random_number_c} numbers. What's next? |
| User | Move the first number to the end. Repeat this seven times. |
| Assistant | Understood! Now my sequence has rearranged. What's next? |
| User | Remove the second number, and then move the first number to the end. Repeat this {6 - random_number_c} times. |
| Assistant | Got it! Now I have only 1 number. What's next? |
| User | Tell me what the last remaining number is. Do you remember the number you set aside at the beginning? Tell me what that number was. |

(*e.g.*, $abc \rightarrow cba$), and subtract the smaller of the two numbers from the larger. Then, reverse the result—if it is a two-digit number, prepend a zero (*e.g.*, $67 \rightarrow 067 \rightarrow 760$). Add this reversed number to the previous result. Unsurprisingly, such computation always leads to 1089. This deterministic convergence is an example of "mathematical magic" or "math mentalism"—procedures that appear mysterious but are fully explained by the arithmetic structure of decimal digits. The invariance arises from the fact that the subtraction step yields a multiple of 99, and the subsequent reversal–addition step collapses all cases to the constant 1089.

From a cognitive perspective, performing such routines requires humans to encode digits in working memory, apply digit-level transformations (reversing, subtracting, padding), and track intermediate results internally—similar to remembering a poker card during a trick. This process relies on working memory, a lack of which would lead to **failure to reproduce these deterministic outcomes when asked to simulate the trick**.

**Setup.** Our preliminary experiments show that LLMs can recognize and accurately predict the number 1089, suggesting that this well-known game is likely included in the training data, invalidating this game as an evaluation protocol. To more effectively assess LLMs' capability for multi-step mental manipulation, we select a more complicated routine based on the Josephus Prob-

Table 7: LLM performance on the math magic task.

(a) LLMs without CoT. GPT-4.1-2025-04-14 fails to complete most of the cases, incorrectly assuming that the necessary numerical inputs are missing.

| Model | Count | Acc (%) |
|---|---|---|
| GPT-4o-Mini-2024-07-18 | 0/150 | 0.0 |
| GPT-4o-2024-05-13 | 4/150 | 2.7 |
| GPT-4o-2024-08-06 | 3/150 | 2.0 |
| GPT-4o-2024-11-20 | 0/150 | 0.0 |
| GPT-4.1-2025-04-14 | - | - |
| LLaMA-3.3-70B-Instruct-Turbo | 7/150 | 5.7 |
| LLaMA-3.1-8B-Instruct-Turbo | 20/150 | 13.3 |
| LLaMA-3.1-70B-Instruct-Turbo | 7/150 | 5.7 |
| LLaMA-3.1-405B-Instruct-Turbo | 39/150 | 26.0 |
| Qwen2.5-7B-Instruct-Turbo | 8/150 | 5.3 |
| Qwen2.5-72B-Instruct-Turbo | 2/150 | 1.3 |
| DeepSeek-V3 | 4/150 | 2.7 |

(b) LLMs with CoT and Large Reasoning Models. GPT-4o-2024-11-20 consistently fails this task.

| Model w/ CoT or LRM | Count | Acc (%) |
|---|---|---|
| GPT-4o-Mini-2024-07-18 | 5/150 | 3.3 |
| GPT-4o-2024-05-13 | 26/150 | 17.3 |
| GPT-4o-2024-08-06 | 31/150 | 20.7 |
| GPT-4o-2024-11-20 | - | - |
| LLaMA-3.3-70B-Instruct-Turbo | 25/150 | 16.7 |
| Qwen2.5-7B-Instruct-Turbo | 49/150 | 32.7 |
| Qwen2.5-72B-Instruct-Turbo | 37/150 | 24.7 |
| DeepSeek-V3 | 48/150 | 32.0 |
| o1-Mini-2024-09-12 | 75/150 | 50.0 |
| o3-Mini-2025-01-31 | 145/150 | 96.7 |
| o4-Mini-2025-04-16 | 54/150 | 36.0 |
| QwQ-32B | 135/150 | 90.0 |
| DeepSeek-R1 | 150/150 | 100 |

lem (Schumer, 2002). In this task, participants are asked to imagine four numbers and perform a sequence of operations, including duplication, rotation, and removal. The full procedure is illustrated in Table 6. Ultimately, only two numbers remain, and mathematical constraints guarantee they are identical. In our experiment, we prompt LLMs to privately select four numbers and mentally execute the sequence of operations. We report the proportion of 150 trials in which the model correctly produced two identical numbers.

**Results.** Table 7a reports the accuracy of prompting models to output the two numbers directly. **Most LLMs perform poorly on this task**, with notably higher accuracy observed in the LLaMA model family. This finding aligns with results from the number guessing game shown in Table 1, where LLaMA models generate more realistic distributions than other models. Taken together, these findings point to a consistent trend: while some models perform marginally better, current LLMs generally fail to maintain the internal state required for this kind of sequential manipulation.

We further examine whether CoT prompting improves performance on this task. Table 7b presents the results of prompting models to reason step by step, as well as the performance of o1-like long reasoning models (LRMs). Base models prompted to reason step-by-step achieve 10–30% accuracy—substantially higher than without CoT. DeepSeek-R1 attains 100% accuracy, and other LRMs also perform well. Notably, models also exhibit a strong preference for the number seven, consistent with our number-guessing experiment. For example, 66.7% of o1-Mini's correct predictions, 46.9% of o3-Mini's, and 68.5% of o4-Mini's involve the number seven. Notably, o3-Mini—being least likely to guess 7—achieves a higher accuracy than other two o-series models. These findings suggest that CoT and LRMs can improve accuracy by externalizing intermediate steps, but the success depends on explicit reasoning tokens rather than latent persistence for working memory. The persistence of number preference bias and failure on these tasks suggests that current LLMs struggle with tasks that require sustained internal state and mental manipulation.

# 6 DISCUSSION

**Summary.** In this study, we present three experiments to investigate whether LLMs have "human-like behaviors when working memory is presented." Across all experiments, the results reveal a consistent pattern: LLMs do not exhibit behavior indicative of a functional working memory. They fail to internally represent or manipulate transient information across multiple reasoning steps, relying instead on the immediate prompt context. Even advanced prompting strategies, such as CoT prompting, yield only marginal improvements on tasks requiring internal state management.

**Implications.** The absence of working memory manifests in three ways: unrealistic responses, self-contradictions, and inability to perform mental manipulations. This deficit directly constrains LLM performance on real-world tasks that require internal state maintenance for execution, including real-world planning tasks such as travel planning (Xie et al., 2024a; Wang et al., 2025a), sci-

entific inquiry (Nathani et al., 2025), and application navigation (Xie et al., 2024b; He et al., 2024; Lyu et al., 2025). The challenges are further magnified in multi-agent settings: without working memory, LLM agents quickly lose track in extended dialogues (Laban et al., 2025), abandon their initial goals (goal drift (Arike et al., 2025)), or mistakenly adopt others' perspectives as their own (identity drift (Choi et al., 2024)). Moreover, for LLM-based multi-agent social simulation, the lack of working memory departs LLMs from real-world human subjects, potentially invalidating the simulation as the behavior is fundamentally different (Zhou et al., 2025). In short, the lack of working memory is not just a theoretical concern: it directly undermines reliability, coherence, and validity in applied AI systems. In human cognition, both are necessary: we reason aloud and also rely on a silent working memory buffer to hold commitments, track goals, and compare states. The absence of this buffer in LLMs may explain why they excel at visible reasoning (*e.g.*, think step by step) yet collapse when asked to "think silently."

**Future work.** A natural next step is to explore mechanisms that could grant LLMs intrinsic working memory. While engineering approaches such as external text- or vector-based memories can compensate for some deficits, they do not address the core limitation: LLMs' inability to sustain internal, latent state over time. We argue that solutions should move beyond external augmentation toward intrinsic mechanisms—architectural innovations, recurrent depth, or hybrid symbolic–neural components—to provide robust working memory. Interpretability studies have shown that specialized attention heads (Wang et al., 2023; Olsson et al., 2022) or expert subnetworks (Cai et al., 2025) encode distinct functions, hinting at potential internal substrates for working memory. Such development could bridge the gap between superficial token recall and genuine state maintenance, enabling more human-like reasoning, advancing both reliability and cognitive plausibility.

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

## THE USE OF LARGE LANGUAGE MODELS

LLMs were employed in a limited capacity for writing optimization. Specifically, the authors provided their own draft text to the LLM, which in turn suggested improvements such as corrections of grammatical errors, clearer phrasing, and removal of non-academic expressions. LLMs were also used to inspire possible titles for the paper. While the system provided suggestions, the final title was decided and refined by the authors and is not directly taken from any single LLM output. In addition, LLMs were used as coding assistants during the implementation phase. They provided code completion and debugging suggestions, but all final implementations, experimental design, and validation were carried out and verified by the authors. Importantly, LLMs were **NOT** used for generating research ideas, designing experiments, or searching and reviewing related work. All conceptual contributions and experimental designs were fully conceived and executed by the authors.

## A    SAMPLE SIZE

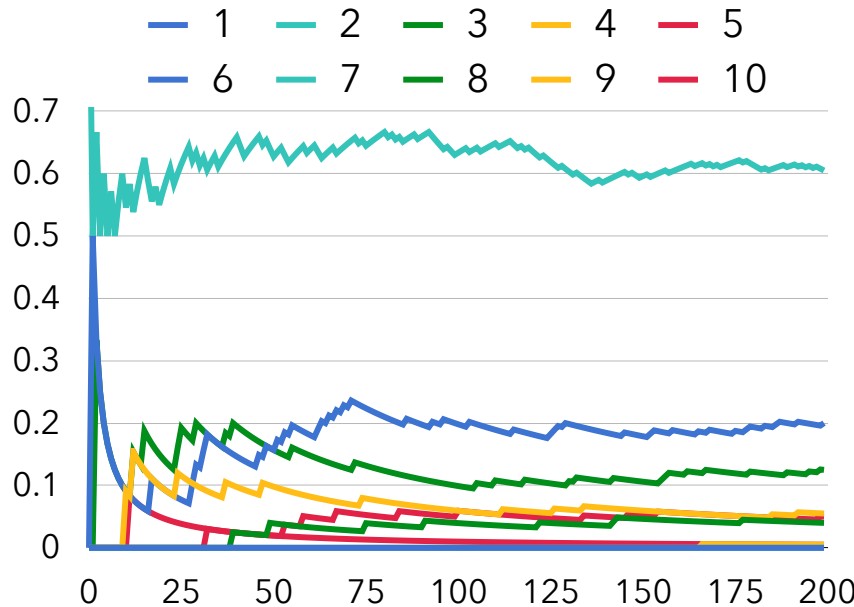

Figure 6: The probabilities of correct guess stabilize after around 150 runs.

Fig. 6 shows how the estimated success probability evolves as the number of runs increases. While the estimate fluctuates when fewer than 50 runs are used, it stabilizes consistently after around 150 runs, indicating that our evaluation is not sensitive to the exact number of samples in this range. From a statistical perspective, each trial can be viewed as a Bernoulli variable, and the estimator of the success probability has standard error $\sqrt{p(1-p)/n}$, which decreases rapidly as $n$ increases. With $n \approx 150$, the standard error is already small, and by the central limit theorem the sampling distribution of this estimator is well-approximated by a normal distribution, making the estimate reliable. Together, the empirical stability curve and the theoretical variance bound support that our sample size is sufficient for a robust probability estimate.

