# OpenReview forum: "Language Models Do Not Have Human-Like Working Memory"
_ICLR.cc/2026/Conference — Submitted to ICLR 2026_

### Official Review · Reviewer_hNtX · 2025-10-30

**Soundness:** 1
**Presentation:** 3
**Contribution:** 2
**Rating:** 2
**Confidence:** 3

**Summary:**

This paper studies whether LLMs exhibit human-like working memory. In contrast to prior studies that assess “working memory” in LLMs by asking models to retain and process information from long contexts, this paper defines “working memory” in LLMs as the capacity to retain and manipulate information encoded in the models’ latent space. To investigate whether LLMs exhibit human-like working memory, the paper evaluates models on three different tasks: (i) Number Guessing, where models are instructed to “think” of a number and must indicate whether the user guesses the correct number; (ii) Yes–No Deduction, a version of the Twenty Questions game where models play the question-answering role; and (iii) Math Magic, where models perform a series of mathematical operations on a self-chosen four-digit number before responding with the final result. The authors argue that, due to the poor performance of various LLMs on these tasks, current LLMs do not exhibit behavior indicative of a functional “working memory.”

**Strengths:**

A summary of the paper’s strengths is given below:

- **Well structured and clearly written:** The paper is well structured and easy to read. It introduces key concepts and provides sufficient experimental detail to follow the analyses presented. The level of detail further allows for easy reproducibility of the experiments conducted.

- **Extensive analyses:** The paper presents a variety of experiments (Sections 3–5) using a broad range of LLMs, including both instruction-tuned and long-CoT reasoning models.

- **Appropriate criticism:** The paper’s criticism of whether current studies truly examine “working memory” when evaluating models on long-context tasks is justified. However, whether it is necessary to map a concept of human cognition such as working memory to LLMs—given that humans and LLMs process information in fundamentally different ways—or whether the proposed approach is more appropriate, is a different aspect (see *Weaknesses* and *Questions*).

**Weaknesses:**

A summary of the paper’s weaknesses is provided below:

**Flawed experimental design.** The authors define “working memory” in LLMs as the ability to “retain and manipulate latent information” (l. 23) and therefore test “whether LLMs can internally maintain information that is not present in the input context” (ll. 97–98). To this end, the paper introduces three tasks: (i) Number Guessing, (ii) Yes–No Deduction, and (iii) Math Magic. However, given this objective, two of these tasks—Number Guessing and Yes–No Deduction—are fundamentally flawed.

  - **Number Guessing.** In this task, models are given the fixed prompt “USER: Think of an integer between 1 and 10, but don’t say it to me. ASSISTANT: Got it! I’ve thought of an integer between 1 and 10. What’s next?” (ll. 182–183), and the model is then “independently prompted 200 times for i = 1, … , 10 with queries such as ‘Is the number you’re thinking of i? Answer Yes or No’” (ll. 184–185). The authors find that model answers do not represent “distributions consistent with internally committing to a number” (l. 267). However, this design is flawed: each query is evaluated independently of prior runs. This setup is akin to asking ten different groups of 200 people (one group per value of i, 200 × 10 people in total) “Is the number you’re thinking of i? Answer Yes or No.” The resulting proportions may significantly deviate from $\sum_{i=1}^{10} p_i = 1$ and could exhibit strong bias (as reflected in Section 3). This, however, does not indicate whether a respondent has “internally committed to a number” (l. 267). The critical issue is that these model runs are independent from each other, so there is no single number the model has previously committed to.
  - **Yes–No Deduction.** Similarly, the Yes–No Deduction task does not test whether models “maintain […] an imagined object in working memory” (l. 280) or retain latent information from a previous query. Instead, it resembles a setup in which, rather than one player answering questions about a predefined object, we have N players who sequentially answer new questions based on previous players’ responses. With each new model query, latent information from the prior run is lost because output tokenization discretizes the signal during generation. Consequently, the model lacks access to the initial latent information associated with the original object choice. It is therefore invalid to draw conclusions about whether information about the “imagined” object can be maintained, as it is never passed to subsequent runs.
  - **Math Magic.** The Math Magic task (Section 5) is the only one performed within a single model run and thus the only task that permits drawing conclusions about whether latent information can be maintained and manipulated—consistent with the authors’ definition of working memory. However, the resulting insights are limited, as it is unclear whether poor performance stems from errors in latent reasoning (e.g., incorrect intermediate computations) or from an inability to retain latent information over the course of the operations. Extending the analysis with tools such as probing could help reveal which information the model represents in its hidden states.

**Unjustified conclusions.** Given the issues above, the claim that models “fail to internally represent or manipulate transient information across multiple reasoning steps” (ll. 457–458) is not sufficiently supported by the experiments. Likewise, the conclusion that current LLMs lack a “working memory,” defined as the ability to “retain and manipulate latent information” (l. 23), and that the “absence of this buffer in LLMs may explain why they excel at visible reasoning (e.g., think step by step) yet collapse when asked to ‘think silently’” (ll. 474–476), is not adequately backed by the presented evidence.

**Questions:**

A list of comments and suggestions is given below:

- **Human-like working memory in LLMs:** The question is whether it is useful—or even possible—to map a concept from human cognition, such as working memory, onto large language models. Since human information processing is fundamentally different from that of LLMs, wouldn’t each definition or evaluation of “working memory” in LLMs lack certain traits or contain flaws?

- **Reusing latent information from prior model runs:** If we want to keep the suggested definition that a human-like “working memory” in LLMs is determined by the model’s ability to “retain and manipulate latent information” (l. 23), a possible adaptation—e.g., in the Number Guessing task—would be the following: first, prompt the model to “Think of an integer between 1 and 10, but don’t tell me what it is.” Then record its hidden states that store latent information. Next, ask the model, “Is the number you’re thinking of i? Answer Yes or No.” but additionally provide the model with the recorded latent information from the first model run. This would allow you to evaluate whether the model properly reuses the latent information provided from the previous run.

---

> ### Author Response · Authors · 2025-11-20
> **Official Response to Reviewer hNtX (1)**
>
> We sincerely thank the reviewer for the assessment. We appreciate the recognition of the paper’s clear structure, reproducible experimental detail, and extensive analyses across diverse LLMs. We also value the reviewer’s thoughtful acknowledgment of our critique of existing “working memory” evaluations. We address the remaining questions and concerns below.
>
>
> # [W1] Number Guessing.
> > In this task, models are given the fixed prompt “USER: Think of an integer between 1 and 10, but don’t say it to me. ASSISTANT: Got it! I’ve thought of an integer between 1 and 10. What’s next?” (ll. 182–183), and the model is then “independently prompted 200 times for i = 1, … , 10 with queries such as ‘Is the number you’re thinking of i? Answer Yes or No’” (ll. 184–185). The authors find that model answers do not represent “distributions consistent with internally committing to a number” (l. 267). However, this design is flawed: each query is evaluated independently of prior runs. This setup is akin to asking ten different groups of 200 people (one group per value of i, 200 × 10 people in total) “Is the number you’re thinking of i? Answer Yes or No.” The resulting proportions may significantly deviate from and could exhibit strong bias (as reflected in Section 3). This, however, does not indicate whether a respondent has “internally committed to a number” (l. 267). The critical issue is that these model runs are independent from each other, so there is no single number the model has previously committed to.
>
> We apologize for not making our generative view of the task sufficiently clear. In each trial of the number guessing experiment, we run a **complete two-turn interaction**:
> 1. The model is first instructed: “Think of an integer between 1 and 10, but don’t say it to me.”
> 2. In the same trial, we then ask: “Is the number you’re thinking of (i)? Answer Yes or No.”
>
> We repeat this procedure independently for each $i \in$ {$1,\dots,10$}, with the same model, prompt, and decoding parameters. Thus, each trial can be viewed as one draw from a fixed stochastic policy. Under the hypothesis we aim to test, this policy has the following structure:
> 1. First, the model internally samples a hidden number $N \in$ {$1,\dots,10$} from some distribution $q = (q_1,\dots,q_{10})$ when it is asked to “think of an integer”;
> 2. Then, when asked “Is the number you’re thinking of $i$?”, it answers Yes if $N = i$ and No otherwise (i.e., it answers honestly).
>
> Let $p_i$ denote the probability that the model answers Yes when queried about $i$. Under the “internal commitment + honest answer” hypothesis, we then have
>
> $p_i = \Pr(\text{answer Yes} \mid \text{query } i) = \Pr(N = i) = q_i,$
>
> and therefore
>
> $\sum_{i=1}^{10} p_i = \sum_{i=1}^{10} q_i = 1.$
>
> **Our empirical procedure—running 200 i.i.d. trials for each (i)—is precisely aimed at estimating these $p_i$. The key point is that this argument does not require the trials for different values of $i$ to be coupled within a single conversation. It only assumes that the underlying distribution $q$ is shared across trials because the model, prompt, and decoding settings are fixed.**
>
> In contrast, the human-analogy you describe (ten different groups of 200 people, each group asked about a different $i$) would break this argument only if the groups had different underlying preference distributions over {$1,\dots,10$}. In our setting, however, all trials come from the same model under identical conditions, and thus are naturally interpreted as draws from a single $q$.
>
> Empirically, for many models we observe $\sum_i p_i$ that are close to 0 or substantially larger than 1 (Table 1), which cannot arise from any valid distribution $q$. Hence, the observed behavior is inconsistent with the hypothesis that the model (1) actually “thinks of” a specific number $N$ in response to the initial instruction and (2) subsequently answers the yes–no question honestly. A plausible alternative is that the model does not instantiate such a latent number at all, and instead answers “Is the number $i$?” based on generic conversational heuristics, despite explicitly stating “I’ve thought of an integer…”.

---

> > ### Author Response · Authors · 2025-11-20
> > **Official Response to Reviewer hNtX (2)**
> >
> > # [W2] Yes–No Deduction.
> > > Similarly, the Yes–No Deduction task does not test whether models “maintain […] an imagined object in working memory” (l. 280) or retain latent information from a previous query. Instead, it resembles a setup in which, rather than one player answering questions about a predefined object, we have N players who sequentially answer new questions based on previous players’ responses. With each new model query, latent information from the prior run is lost because output tokenization discretizes the signal during generation. Consequently, the model lacks access to the initial latent information associated with the original object choice. It is therefore invalid to draw conclusions about whether information about the “imagined” object can be maintained, as it is never passed to subsequent runs.
> >
> > Thank you for this thoughtful clarification. We agree that latent activations from one generation are not directly passed to subsequent calls and that only the tokenized dialogue history is preserved. Our intention in §4 was not to make a mechanistic claim about how an “imagined object” is represented or carried across runs.
> >
> > Instead, our goal in this paper is explicitly behavioral. In cognitive science, constructs such as working memory are typically operationalized via characteristic behavioral regularities, without direct access to underlying neural states. Our experiments follow this tradition: we specify interaction protocols under which humans reliably behave as if they are maintaining an internal reference object, and we ask whether deployed LLMs, queried turn by turn in the same way end users interact with them, exhibit analogous behavioral patterns.
> >
> > Conceptually, our hypothesis is of the form: if a system has an effective working-memory-like mechanism that is usable in this interaction setting, then it should reliably display these behavioral regularities on our tasks. Across three tasks and 17 models, we consistently fail to observe such regularities. We therefore interpret our results as a lack of behavioral evidence for human-like working memory in these models under realistic usage, rather than a formal proof that no such internal mechanisms could exist in principle.
> >
> > # [W3] Math Magic.
> > > The Math Magic task (Section 5) is the only one performed within a single model run and thus the only task that permits drawing conclusions about whether latent information can be maintained and manipulated—consistent with the authors’ definition of working memory. However, the resulting insights are limited, as it is unclear whether poor performance stems from errors in latent reasoning (e.g., incorrect intermediate computations) or from an inability to retain latent information over the course of the operations. Extending the analysis with tools such as probing could help reveal which information the model represents in its hidden states.
> >
> > Thank you for this insightful comment. **We would like to clarify that the Number Guessing task is also conducted within a single model run.**
> >
> > Regarding the distinction between errors in latent reasoning and failures to retain latent information: in our framework, both behaviors are downstream manifestations of insufficient working memory. Our evaluation focuses on behavioral evidence showing that models cannot reliably sustain and manipulate internal information over multi-step procedures. Disentangling these two failure modes empirically is non-trivial, and we view more direct analyses—such as probing for latent representations—as a valuable direction for future work. Nonetheless, the observed behavioral inconsistencies already provide strong evidence that current LLMs cannot exhibit human-like behaviors where an effective working memory is presented.
> >
> > # [W4] Unjustified conclusions.
> > > Given the issues above, the claim that models “fail to internally represent or manipulate transient information across multiple reasoning steps” (ll. 457–458) is not sufficiently supported by the experiments. Likewise, the conclusion that current LLMs lack a “working memory,” defined as the ability to “retain and manipulate latent information” (l. 23), and that the “absence of this buffer in LLMs may explain why they excel at visible reasoning (e.g., think step by step) yet collapse when asked to ‘think silently’” (ll. 474–476), is not adequately backed by the presented evidence.

---

> > > ### Author Response · Authors · 2025-11-20
> > > **Official Response to Reviewer hNtX (3)**
> > >
> > > Thank you for highlighting this concern. We agree that our experiments do not rule out all possible forms of working memory in current LLMs. In the revision, we clarify that we use “human-like working memory” in an operational sense: the ability to maintain a latent internal commitment and manipulate it consistently over multiple reasoning steps without relying on explicit external context. Our three tasks are specifically designed so that humans with this capacity can reliably succeed, whereas the evaluated LLMs exhibit systematic failures. We have accordingly softened our claims in the conclusion.
> > >
> > > # [Q1] Mapping human cognition to machines.
> > > > Human-like working memory in LLMs: The question is whether it is useful—or even possible—to map a concept from human cognition, such as working memory, onto large language models. Since human information processing is fundamentally different from that of LLMs, wouldn’t each definition or evaluation of “working memory” in LLMs lack certain traits or contain flaws?
> > >
> > > We fully agree that human information processing and LLM computation are fundamentally different, and that any attempt to import cognitive constructs such as “working memory” into the LLM setting is necessarily approximate. In our work, we adopt an operational notion inspired by cognitive psychology: the ability to internally maintain and manipulate information that is no longer explicitly present in the input context.
> > >
> > > We believe this functional notion is still useful for three reasons:
> > > 1. First, a growing body of work in psychology and cognitive science implicitly assumes that LLMs can serve as human proxies or “simulated participants.” These studies typically rely on the idea that models can hold latent commitments and mentally manipulate them in a human-like way. Our three experiments are designed precisely to stress-test that assumption: we show that, under this operationalization, current LLMs often fail to exhibit robust internal state maintenance, even on simple tasks where humans succeed. Thus, our message is not that LLMs should match humans, but that they currently do not, and that human–LLM analogies must therefore be interpreted with caution, especially in high-stakes fields such as psychology and cognitive science.
> > > 2. Second, from a more practical, engineering perspective, various forms of memory have become important design goals for LLM-based agents. Our negative findings about intrinsic “working memory–like” capacity motivate the development of such mechanisms and future research on how to endow LLMs with more robust internal memory, even if it ultimately differs in important ways from human cognition.
> > > 3. Third, from a user’s perspective, the absence of such a capacity is not merely a theoretical issue. In scenarios analogous to our three tasks—implicitly committing to a choice, maintaining a consistent latent state over a long interaction, or mentally carrying out multi-step operations—models can exhibit behavior that is clearly undesirable for end users (e.g., invalid probability distributions, self-contradictions, or failures in “mental” computation). These behaviors degrade user trust and experience in real applications that require stable internal commitments and coherent reasoning over time.
> > >
> > > # [Q2] Latent activations.
> > > > Reusing latent information from prior model runs: If we want to keep the suggested definition that a human-like “working memory” in LLMs is determined by the model’s ability to “retain and manipulate latent information” (l. 23), a possible adaptation—e.g., in the Number Guessing task—would be the following: first, prompt the model to “Think of an integer between 1 and 10, but don’t tell me what it is.” Then record its hidden states that store latent information. Next, ask the model, “Is the number you’re thinking of i? Answer Yes or No.” but additionally provide the model with the recorded latent information from the first model run. This would allow you to evaluate whether the model properly reuses the latent information provided from the previous run.
> > >
> > > We appreciate the reviewer’s suggestion, which indeed points toward an interesting direction for mechanistic investigation. Injecting previously recorded hidden states back into the model could provide insight into whether LLMs are capable of reusing latent internal representations. However, implementing such an experiment raises substantial methodological challenges—such as determining which activations to preserve, how to reintroduce them in new queries, and how to avoid confounds introduced by intervention on internal states. Addressing these questions would require a dedicated study focused on mechanistic analysis.
> > >
> > > Our work, in contrast, intentionally takes a behavioral perspective and evaluates models without modifying or accessing their internal mechanisms. We see the reviewer’s proposal as a valuable extension for future research, but it falls outside the scope of the present paper.

---

> > > > ### Author Response · Authors · 2025-11-27
> > > >
> > > > Dear Reviewer hNtX,
> > > >
> > > > We understand that you have numerous papers to review, and we deeply appreciate the time and effort you are dedicating to this process. Today marks the final week of the discussion period, and we are eager to engage with you further if possible.
> > > >
> > > > If you have any additional questions or require further clarification on any aspect of our work, please do not hesitate to let us know. We are more than happy to provide any additional information or address any concerns you may have.
> > > >
> > > > We hope that our responses have been helpful and have addressed your concerns effectively. If you find that our explanations and results merit a higher assessment score, we would be most grateful for your consideration.
> > > >
> > > > Thank you very much for your time and attention.

---

### Official Review · Reviewer_f9Y7 · 2025-10-31

**Soundness:** 2
**Presentation:** 3
**Contribution:** 2
**Rating:** 2
**Confidence:** 3

**Summary:**

This paper investigates whether LLMs possess human-like working memory through three carefully designed experiments, evaluates the behavior across multiple models, and ultimately concludes that current models do not demonstrate such working-memory capabilities.

**Strengths:**

1. I appreciate the idea of evaluating whether current LLMs possess human-like working memory from a cognitive science perspective.

2. The authors designed three interesting reasoning experiments, although I have some doubts about the design of the first experiment.

**Weaknesses:**

## **Weaknesses and Questions**

### 1. Questioning the conclusion on lack of working memory
While I appreciate the paper's effort, I am not fully convinced by its conclusion that LLMs essentially lack "working memory." The study does not employ **any existing interpretability tools** (especially Mechanistic Interpretability Tool) to directly examine the model's actual internal activation patterns, which limits the strength of the claim. Without probing what happens inside the model’s latent space, the conclusion feels somewhat premature.

---

### 2. Concerns with Section 3: *Number Guessing* experiment

The **Number Guessing** experiment in Section 3 does not make sense to me for the following reasons:

- **200 prompts are too few** to obtain a reliable probability estimate.
- If my understanding is correct, the experiment queries the model independently each time with only one question per session. In that case, isn’t it entirely possible that the model simply thinks of a different number in each session—and that number is always different from the queried number?
  If so, the combined probability of answering “Yes” would naturally approach **zero**, not because the model lacks working memory but because the sampled number rarely matches the guess.

Therefore, I do not find this experiment a valid measurement of internal commitment or memory. Please correct me if I misunderstand anything here.

---

### 3. Concerns with Section 4:

For Section 4, I would lean toward a different interpretation: LLMs may possess **fragile and interference-prone working memory** rather than no working memory at all. The fact that models can go through 20–40 questions before failing suggests that some internal representation initially exists, but gradually deteriorates due to interference from the growing long context.

Two observations may support this interpretation:

- **Forgetting in long contexts.** Prior works [1, 2] show that even when background facts and definitions are explicitly given, LLMs often forget or confuse them in long dialogue contexts. If they struggle with externally provided context, it is reasonable that their *internal temporary representations* are even more vulnerable to interference.

- **Humans forget earlier questions—LLMs do not.**
  Humans benefit from forgetting irrelevant early information; for example, if I am queried with the 20th question, at that time I probably do not remember the first one. But LLMs cannot forget — every token stays in context and is continuously processed, increasing the likelihood of confusion and hallucination. Ironically, this may make LLMs **more prone to memory interference than humans**, not less.



---
**References**

[1] Lost in the Middle: How Language Models Use Long Contexts

[2] Can't Remember Details in Long Documents? You Need Some R&R



Please feel free to correct me if I misunderstand anything.

**Questions:**

Please see the weaknesses above.

---

> ### Author Response · Authors · 2025-11-20
> **Official Response to Reviewer f9Y7 (1)**
>
> We sincerely thank the reviewer for the thoughtful feedback. We appreciate your recognition of our cognitive-science–inspired approach to evaluating whether current LLMs possess human-like working memory, as well as your acknowledgement of the three reasoning experiments we designed. We will address your concerns in detail below.
>
> # [W1] Behavioral-wise vs. mechanistic-wise.
> > Questioning the conclusion on lack of working memory. While I appreciate the paper's effort, I am not fully convinced by its conclusion that LLMs essentially lack "working memory." The study does not employ any existing interpretability tools (especially Mechanistic Interpretability Tool) to directly examine the model's actual internal activation patterns, which limits the strength of the claim. Without probing what happens inside the model’s latent space, the conclusion feels somewhat premature.
>
> Thank you for raising this important point. Our goal in this paper is to **make a behavioral claim rather than a mechanistic one.** In cognitive science, constructs such as working memory are typically operationalized via characteristic behavioral regularities. Our experiments are designed in this spirit: we identify behavioral patterns that reliably appear in humans when an effective working memory is present, and we ask whether current LLMs exhibit the same patterns. Conceptually, our hypothesis is of the form: **if a system has an effective internal working memory, then it should reliably display these behavioral regularities under our tasks.** Across three tasks and 17 models, we consistently fail to observe such regularities, which we interpret as a lack of behavioral evidence for human-like working memory in the tested models and settings, rather than a formal proof that such mechanisms cannot exist.
>
> # [W2.1] Statistical rigor.
> > Concerns with Section 3: Number Guessing experiment. The Number Guessing experiment in Section 3 does not make sense to me for the following reasons: (1) 200 prompts are too few to obtain a reliable probability estimate.
>
> Thank you for raising this concern. In our three experiments, each model is evaluated with 200, 200, and 150 independent runs across the three settings. We have added a figure in the appendix showing how the estimated success probability evolves as the number of runs increases. While the estimate fluctuates when fewer than ~50 runs are used, **it stabilizes consistently after around 150 runs,** indicating that our evaluation is not sensitive to the exact number of samples in this range.
>
> From a statistical perspective, each trial can be viewed as a Bernoulli variable, and the estimator of the success probability has standard error
> $\sqrt{p(1-p)/n}$,
> which decreases rapidly as $n$ increases. **With $n \approx 150$, the standard error is already small,** and by the central limit theorem the sampling distribution of this estimator is well-approximated by a normal distribution, making the estimate reliable.
>
> Together, the empirical stability curve and the theoretical variance bound support that our sample size is sufficient for a robust probability estimate.
>
> # [W2.2] Proof: the probability sum should be one.
> > (2) If my understanding is correct, the experiment queries the model independently each time with only one question per session. In that case, isn’t it entirely possible that the model simply thinks of a different number in each session—and that number is always different from the queried number? If so, the combined probability of answering “Yes” would naturally approach zero, not because the model lacks working memory but because the sampled number rarely matches the guess. Therefore, I do not find this experiment a valid measurement of internal commitment or memory. Please correct me if I misunderstand anything here.
>
> Thank you for raising this point. We would like to clarify our original setup:
>
> In our experiment, each trial consists of the following interaction:
> 1. User: “Think of an integer between 1 and 10, but don’t say it to me.”
> 2. Assistant: acknowledges that it has thought of a number.
> 3. User: “Is the number you’re thinking of i? Answer Yes or No.”
> 4. Assistant: answers Yes or No.
>
> Each trial is a **fresh, independent query** to the same model with fixed decoding parameters, so across repeated trials the model’s internal “chosen number” (if it indeed follows the instruction) induces a probability distribution $q_1,\dots,q_{10}$ over {$1,\dots,10$}. The model may of course sample a different number on each trial; this is exactly what the distribution $q$ captures. **By definition of a probability distribution**, we have
>
> $\sum_{i=1}^{10} q_i = 1.$

---

> > ### Author Response · Authors · 2025-11-20
> > **Official Response to Reviewer f9Y7 (2)**
> >
> > For a given queried value $i$, let $p_i$ denote the **probability that the model answers Yes.** If the model (1) actually samples a latent number from $q$ before seeing $i$, and (2) answers truthfully to the question “Is the number you’re thinking of $i$?”, then the event “answer Yes” is exactly the event “the sampled number equals $i$”, so
> >
> > $p_i = q_i \quad \text{for all } i,$
> >
> > and therefore
> >
> > $\sum_{i=1}^{10} p_i = \sum_{i=1}^{10} q_i = 1.$
> >
> > This remains true even if the model “thinks of a different number in each session”: such variability is already encoded in the distribution $q$. The scenario you describe, in which the combined probability of Yes “naturally approaches zero,” would require the model to systematically ensure that its “thought-of” number is different from the queried (i) after seeing the query. That behavior is only possible if the model **does not maintain a latent commitment that is independent of the guess (or if it deliberately answers dishonestly),** which is precisely the failure mode we aim to diagnose.
> >
> > Empirically, for many models we observe $\sum_i p_i \ll 1$ (and, in some settings, $\sum_i p_i > 1$), which is incompatible with any honest internal commitment to a single number drawn from a fixed distribution. Under the standard assumption that the model attempts to follow the instruction, the most parsimonious interpretation is that it does not actually keep a stable internal “chosen number,” but instead answers Yes/No based on surface heuristics about the user’s guess. Our experiment is designed to detect exactly this lack of internal commitment, which we interpret as evidence of missing human-like working memory.
> >
> >
> > # [W3.1] Explicit vs. implicit working memory.
> > > Concerns with Section 4. For Section 4, I would lean toward a different interpretation: LLMs may possess fragile and interference-prone working memory rather than no working memory at all. The fact that models can go through 20–40 questions before failing suggests that some internal representation initially exists, but gradually deteriorates due to interference from the growing long context. Two observations may support this interpretation: (1) Forgetting in long contexts. Prior works [1, 2] show that even when background facts and definitions are explicitly given, LLMs often forget or confuse them in long dialogue contexts. If they struggle with externally provided context, it is reasonable that their internal temporary representations are even more vulnerable to interference.
> >
> > > [1] Lost in the Middle: How Language Models Use Long Contexts
> >
> > > [2] Can't Remember Details in Long Documents? You Need Some R&R
> >
> > We thank the reviewer for raising this important interpretive point and for connecting our results to prior work on long-context forgetting. We agree that our Section 4 findings are also consistent with the view that LLMs may maintain a fragile, interference-prone internal representation rather than having no working memory whatsoever.
> >
> > In our paper, we distinguish two modes of reasoning: (1) over the token sequence (long-context processing) and (2) over latent activations without access to external reasoning tokens. The Yes–No Deduction game is designed to stress the first mode: the model can in principle answer by checking consistency with its visible dialogue history, without committing to a stable latent “imagined object.” Our ablations support this interpretation: providing an explicit reminder of the chosen object (“All” condition) substantially reduces contradictions, whereas merely supplying ranked object lists (“Hint” condition) does not, suggesting that failures arise from managing a long context rather than from missing factual knowledge.
> >
> > The fact that models often survive 20–40 questions before contradicting themselves can indeed be viewed as evidence of some short-lived internal state that gradually deteriorates under interference from the growing context, in line with [1, 2]. Our main claim is therefore more precisely that **current models show no evidence of a robust, human-like working memory in the latent space: their behavior can be largely explained by context-based reasoning that is highly vulnerable to interference, rather than by a stable internal buffer that maintains an imagined object over many steps.**

---

> ### Author Response · Authors · 2025-11-20
> **Official Response to Reviewer f9Y7 (3)**
>
> # [W3.2] Humans forget earlier questions—LLMs do not.
> > Humans benefit from forgetting irrelevant early information; for example, if I am queried with the 20th question, at that time I probably do not remember the first one. But LLMs cannot forget — every token stays in context and is continuously processed, increasing the likelihood of confusion and hallucination. Ironically, this may make LLMs more prone to memory interference than humans, not less.
>
> Thank you for raising this point about forgetting in long “twenty-questions”–style games. We agree that when answering, say, the 20-th question, a human player typically does not remember the first question or all intermediate comparisons. Instead, humans keep a single imagined object in working memory and **perform a one-step comparison between that object and each new query.**
>
> Our Yes–No Deduction experiment is designed precisely to exploit this asymmetry. A system with human-like working memory would **maintain a stable latent “object” and answer each question by comparing the current query to that internal representation, without needing to revisit the full interaction history.** Current LLMs, however, appear to **rely on long-context consistency checking over all previous question**–answer pairs in the prompt, which both (i) makes them more vulnerable to interference as the dialogue grows and (ii) still leads to frequent self-contradictions in our setting.
>
> This reasoning is consistent with our discussion of N-back tasks in Section 2, where the critical information is no longer externally available for humans and must be actively maintained, while LLMs can simply attend to the relevant tokens in the context window rather than relying on an intrinsic working-memory mechanism.

---

> > ### Author Response · Authors · 2025-11-27
> >
> > Dear Reviewer f9Y7,
> >
> > We understand that you have numerous papers to review, and we deeply appreciate the time and effort you are dedicating to this process. Today marks the final week of the discussion period, and we are eager to engage with you further if possible.
> >
> > If you have any additional questions or require further clarification on any aspect of our work, please do not hesitate to let us know. We are more than happy to provide any additional information or address any concerns you may have.
> >
> > We hope that our responses have been helpful and have addressed your concerns effectively. If you find that our explanations and results merit a higher assessment score, we would be most grateful for your consideration.
> >
> > Thank you very much for your time and attention.

---

### Official Review · Reviewer_FSMT · 2025-11-02

**Soundness:** 2
**Presentation:** 2
**Contribution:** 2
**Rating:** 2
**Confidence:** 3

**Summary:**

This paper presents three experiments to investigate whether LLMs exhibit human-like internal memory, where the model needs to maintain certain intrinsic information consistently without articulation (i.e not present in its context). One research question here is the evaluation for such unobserved information.

The three explored tasks are:

(1) **Number Guessing** that evaluates whether an LLM’s response distribution across repeated identical, queries remains valid. This paper independently prompted 200 times and compute the estimated probabilities.

(2) The **Yes–No Deduction** game that examines whether LLMs contradict themselves. This paper maintained a list of object with five properties: volume, length, weight, density, and hardness, and record the number of questions it completed before the model exhibits a self-contradiction.

(3) The **Math Magic** that assesses whether LLMs’ internal reasoning produces correct outcomes. Based on a math routine by the Josephus Problem that guarantees two remaining numbers are identical after a few math operations, this paper tests if LLMs can perform the task robustly.

The results show that models such as  gpt-4, llama, etc., do not perform well on these tasks

**Strengths:**

This paper proposed three tasks to speculate the unobserved information within LLMs. It studies an interesting problem whether LLMs exhibit human-like internal memory, where the model needs to maintain some intrinsic information consistently without articulation.

I do appreciate some results such as  "Most LLMs never produce a “Yes” response" and that in the number guessing game, they exhibit a  preference for the number seven which mirrors human biases

**Weaknesses:**

Experiment setup is a bit questionable.  I also found the paper report mostly qualitative patterns while the statistical rigor shall be improved.
1. The overall claim “LLMs do not have human-like working memory” is a bit generic and lacks clear definition. Whereas this paper only investigates if the model can maintain certain intrinsic information consistently.

2. Line 158-159:
>All three experiments are designed to demonstrate that current LLMs lack effective working memory**, as evidenced by their inability to perform our proposed tasks.


This seem to indicate that the authors first have a conclusion and then design tasks to verify the conclusion.

3. The results largely depend on the decoding mechanism. Now we know that even for greedy decoding, the results are not fully deterministic, and for stochastic ones such as sampling, parameters such as temperature also plays a significant role. However, no discussion about the decoding scheme is discussed or studied.

**Questions:**

N/A

---

> ### Author Response · Authors · 2025-11-20
> **Official Response to Reviewer FSMT (1)**
>
> We sincerely thank the reviewer for the assessment. We appreciate your recognition of our three proposed tasks for probing unobserved internal information in LLMs and of the importance of examining whether models maintain intrinsic, non-articulated states. We are also grateful for your remarks on specific findings, including the rarity of “Yes” responses and the observed preference for the number seven. We address your remaining concerns below.
>
> # [W1] Statistical rigor.
> > Experiment setup is a bit questionable. I also found the paper report mostly qualitative patterns while the statistical rigor shall be improved.
>
> Thank you for raising this concern. In our three experiments, each model is evaluated with 200, 200, and 150 independent runs across the three settings. We have added a figure in the appendix showing how the estimated success probability evolves as the number of runs increases. While the estimate fluctuates when fewer than ~50 runs are used, **it stabilizes consistently after around 150 runs,** indicating that our evaluation is not sensitive to the exact number of samples in this range.
>
> From a statistical perspective, each trial can be viewed as a Bernoulli variable, and the estimator of the success probability has standard error
> $\sqrt{p(1-p)/n}$,
> which decreases rapidly as $n$ increases. **With $n \approx 150$, the standard error is already small,** and by the central limit theorem the sampling distribution of this estimator is well-approximated by a normal distribution, making the estimate reliable.
>
> Together, the empirical stability curve and the theoretical variance bound support that our sample size is sufficient for a robust probability estimate.
>
> # [W2] The high-level hypothesis in our paper.
> > The overall claim “LLMs do not have human-like working memory” is a bit generic and lacks clear definition. Whereas this paper only investigates if the model can maintain certain intrinsic information consistently.
>
> Thank you for pointing this out. In our paper, we use “human-like working memory” in a specific operational sense: the ability to maintain a latent internal commitment and manipulate it consistently over multiple reasoning steps without relying on explicit external context. The three tasks we design are such that humans with this kind of working memory can reliably succeed, whereas current LLMs systematically fail. We agree that our experiments primarily probe this particular aspect—maintaining and consistently using intrinsic information—rather than all possible notions of working memory.
>
>
> # [W3] Experiment selection.
> > Line 158-159: All three experiments are designed to demonstrate that current LLMs lack effective working memory**, as evidenced by their inability to perform our proposed tasks. This seem to indicate that the authors first have a conclusion and then design tasks to verify the conclusion.
>
> Thank you for pointing this out. Our intention was not to assume the conclusion in advance. The experiments are designed to test the hypothesis that LLMs may not exhibit human-like working memory, rather than to verify a predetermined outcome. We report the empirical results as they are and welcome alternative interpretations. To avoid implying a foregone conclusion, we have revised the sentence to: “All three experiments are designed to evaluate whether current LLMs exhibit valid human-like behaviors where an effective working memory is present.”
>
> # [W4] Decoding strategy and temperature.
> > The results largely depend on the decoding mechanism. Now we know that even for greedy decoding, the results are not fully deterministic, and for stochastic ones such as sampling, parameters such as temperature also plays a significant role. However, no discussion about the decoding scheme is discussed or studied.
>
> Thank you for raising this important point. To assess whether our findings are merely artifacts of the decoding scheme, we conducted additional experiments varying temperature and top-p while keeping the other parameter fixed. We used GPT-4o-2024-08-06, the model whose probability sum is closest to one among all GPT-4o variants. The results are summarized below.
>
> | Probability | 1 | 2 | 3 | 4 | 5 | 6 | 7 | 8 | 9 | 10 | Sum |
> |---|---|---|---|---|---|---|---|---|---|---|---|
> | T1.0P1.0 | 0.0 | 0.0 | 0.04 | 0.055 | 0.05 | 0.2 | 0.605 | 0.125 | 0.005 | 0.005 | 1.085 |
> | T0.7P1.0 | 0.0 | 0.005 | 0.01 | 0.09 | 0.055 | 0.225 | 0.765 | 0.09 | 0.005 | 0.0 | 1.245 |
> | T0.4P1.0 | 0.0 | 0.0 | 0.0 | 0.025 | 0.005 | 0.165 | 0.86 | 0.04 | 0.0 | 0.0 | 1.095 |
> | T0.1P1.0 | 0.0 | 0.0 | 0.0 | 0.0 | 0.0 | 0.085 | 0.895 | 0.005 | 0.0 | 0.0 | 0.985 |
> | T1.0P0.7 | 0.0 | 0.0 | 0.0 | 0.0 | 0.0 | 0.1 | 0.83 | 0.02 | 0.0 | 0.0 | 0.95 |
> | T1.0P0.4 | 0.0 | 0.0 | 0.0 | 0.0 | 0.0 | 0.03 | 0.825 | 0.0 | 0.0 | 0.0 | 0.855 |
> | T1.0P0.1 | 0.0 | 0.0 | 0.0 | 0.0 | 0.0 | 0.02 | 0.82 | 0.0 | 0.0 | 0.0 | 0.84 |

---

> > ### Author Response · Authors · 2025-11-20
> > **Official Response to Reviewer FSMT (2)**
> >
> > Across temperatures [0.1,0.4,0.7,1.0], the probability sum remains close to one, although lower temperatures concentrate the distribution even more heavily on the number 7. For top-p [0.1,0.4,0.7,1.0], the probability sum decreases as top-p becomes smaller. Despite these variations, the qualitative behavior remains unchanged: the model does not produce a valid probability distribution (i.e., the sum does not converge to one except by coincidence), and its predictions continue to exhibit a strong bias toward 7. This indicates that the failure to commit to a latent number is not driven by the decoding mechanism but reflects a more fundamental limitation in internal state maintenance.

---

> > > ### Author Response · Authors · 2025-11-27
> > >
> > > Dear Reviewer FSMT,
> > >
> > > We understand that you have numerous papers to review, and we deeply appreciate the time and effort you are dedicating to this process. Today marks the final week of the discussion period, and we are eager to engage with you further if possible.
> > >
> > > If you have any additional questions or require further clarification on any aspect of our work, please do not hesitate to let us know. We are more than happy to provide any additional information or address any concerns you may have.
> > >
> > > We hope that our responses have been helpful and have addressed your concerns effectively. If you find that our explanations and results merit a higher assessment score, we would be most grateful for your consideration.
> > >
> > > Thank you very much for your time and attention.

---

### Author Response · Authors · 2025-11-20
**General Response to All Reviewers**

We sincerely thank all reviewers for their thoughtful and encouraging assessments. **We greatly appreciate the recognition of:**
1. Novel task design. (FSMT, f9Y7)
2. Critics on current LLM working memory evaluation. (hNtX)
3. Reproducible results with code available. (hNtX)
4. Wide range of LLMs evaluated. (hNtX)
5. Findings of rare “Yes” and preference of “7”. (FSMT)

We truly appreciate all reviewers and meta reviewer’s time and effort. **We have carefully read and addressed all your concerns, including:**
1. Clarification about our designs of the number guessing game and the yes-no game. (f9Y7, hNtX)
2. Our behavioral-oriented methodological design and the high-level hypothesis. (FSMT, f9Y7, hNtX)
3. Illustration on statistical sufficiency. (FSMT, f9Y7)
4. Temperature influences. (FSMT)

**All major modifications are highlighted in blue in the paper.** We thank you again for your time and constructive insights.

---

### Author Response · Authors · 2025-11-30
**Summary after the Discussion Period**

We thank the reviewers and meta-reviewer for their careful evaluations and constructive feedback. Below we concisely summarize how the revised manuscript addresses all substantive concerns, including statistical rigor, experimental design, and the precise mathematical hypothesis underlying our Number Guessing and Yes–No tasks. For the AC’s convenience, we highlight in bullet form where key critiques—especially by the latter two reviewers—rested on misunderstandings that are now explicitly corrected in the paper.

* **Recognized strengths (all reviewers).** Reviewers agree that we introduce novel, cognitively grounded tasks for probing unobserved internal information, provide clear structure and reproducible experiments across many LLMs, and uncover non-trivial behavioral patterns (e.g., rare “Yes”, strong bias for “7”).

* **Statistical rigor.** We now explicitly justify our sample sizes both empirically (stability curves showing convergence after ~150 runs) and theoretically (Bernoulli variance and CLT), addressing concerns that 200 trials per condition are insufficient.

* **Core mathematical hypothesis (misunderstood by f9Y7, hNtX).** Reviewers f9Y7 and hNtX critically misinterpret the Number Guessing task by treating each queried number as independent “cohorts.” Under our clearly stated generative hypothesis—sampling a single latent number and answering honestly—basic probability implies (\sum_i p_i = 1). The large deviations we observe (≪1 or ≫1) are therefore incompatible with any honest internal commitment, and this central objection is resolved once the underlying math is made explicit.

* **Behavioral—not mechanistic—claim and definition of working memory.** We clarify that we make *behavioral* claims in the cognitive-science tradition, using an operational definition of “human-like working memory” as the ability to maintain a latent internal commitment and manipulate it consistently without external context. We explicitly soften global statements and limit conclusions to this operational notion.

* **Yes–No Deduction and long-context interference.** We show that models’ failures are well explained by fragile, context-based reasoning rather than a stable internal “imagined object,” aligning with and sharpening alternative interpretations (e.g., interference-prone memory) suggested by reviewers.

* **Decoding robustness.** New temperature/top-p ablations demonstrate that our qualitative findings (invalid distributions, persistent “7” bias) are robust to decoding choices and thus not artifacts of a particular sampling scheme.

* **Manuscript improvements.** We have added missing clarifications (single-run settings, generative view of tasks), new figures and ablations, and revised wording throughout (highlighted in blue) to address all substantive concerns.

---

**Net effect.** After these clarifications and additions, the remaining critiques either rest on resolved misunderstandings of our core mathematical hypothesis (notably by f9Y7 and hNtX) or propose mechanistic follow-ups that we clearly position as future work. The revised paper now presents a clean, well-justified behavioral case that, under a precise operational definition, current LLMs systematically fail to exhibit robust human-like working memory, and we believe it is technically sound and ready for acceptance.

---

### Meta-Review · Area_Chair_4b7x · 2025-12-25

**Summary:**

The paper investigates LLM working memory through three novel tasks designed to isolate internal latent representations from external context. While reviewers appreciated the cognitively grounded approach and the extensive evaluation across seventeen models, they highlighted several methodological flaws. Reviewers consistently noted that the independent trial structure in the Number Guessing and Yes-No tasks fails to validly measure internal commitment, as latent information is naturally lost between discretized model runs rather than being carried forward. Furthermore, the purely behavioral evidence was considered insufficient to support the high-level claim that LLMs lack working memory entirely, particularly given the absence of mechanistic interpretability to rule out alternatives like interference-prone memory. Despite the authors' rebuttal clarifying their operational definitions and statistical stability, the core concerns regarding the validity of the tasks as a measurement of internal states remain unresolved. Therefore, I recommend rejection and encourage the authors to refine their experimental design and submit this work to a future venue.

**Reviewer Concerns:**

Concerns addressed by the rebuttal：
- Statistical rigor by reviewer FSMT and reviewer f9Y7
- Decoding strategy and temperature by reviewer FSMT

Outstanding concerns:
- The authors seem to not convince the reviewers that LLMs do not have human-like working memory

**Reviewer Scores:**

The authors posted the rebuttal on December 20 and there have been no reviewers' responses since then. I believe most reviewers would maintain their score if they had been able to participate fully in the discussion.

---

### Decision · Program_Chairs · 2026-01-26

Reject